# A Case Study of the Morphological and Molecular Variation within a Ciliate Genus: Taxonomic Descriptions of Three *Dysteria* Species (Ciliophora, Cyrtophoria), with the Establishment of a New Species

**DOI:** 10.3390/ijms23031764

**Published:** 2022-02-03

**Authors:** Xuetong Zhao, Hui Zhang, Qianqian Zhang, Zhishuai Qu, Alan Warren, Di Wu, Xiangrui Chen

**Affiliations:** 1School of Marine Sciences, Ningbo University, Ningbo 315800, China; zhaoxuetong1102@163.com (X.Z.); zzhang_hui@163.com (H.Z.); wudizjnb@163.com (D.W.); 2Yantai Institute of Coastal Zone Research, Chinese Academy of Sciences, Yantai 264003, China; qqzhang@yic.ac.cn; 3State Key Laboratory of Marine Environmental Science, College of the Environment & Ecology, Xiamen University, Xiamen 361104, China; quzhishuai@126.com; 4Department of Life Sciences, Natural History Museum, London SW7 5BD, UK; a.warren@nhm.ac.uk

**Keywords:** ciliature, molecular phylogeny, morphological diversity, new species, SSU rDNA sequence

## Abstract

Three *Dysteria* species, *D. crassipes* Claparède & Lachmann, 1859; *D. brasiliensis* Faria et al., 1922; and *D. paracrassipes* n. sp., were collected from subtropical coastal waters of the East China Sea, near Ningbo, China. The three species were studied based on their living morphology, infraciliature, and molecular data. The new species *D. paracrassipes* n. sp. is very similar to *D. crassipes* in most morphological features except the preoral kinety, which is double-rowed in the new species (vs. single-rowed in *D. crassipes*). The difference in the small ribosomal subunit sequences (SSU rDNA) between these two species is 56 bases, supporting the establishment of the new species. The Ningbo population of *D. crassipes* is highly similar in morphology to other known populations. Nevertheless, the SSU rDNA sequences of these populations are very different, indicating high genetic diversity and potentially cryptic species. *Dysteria brasiliensis* is cosmopolitan with many described populations worldwide and four deposited SSU rDNA sequences. The present work supplies morphological and molecular information from five subtropical populations of *D. brasiliensis* that bear identical molecular sequences but show significant morphological differences. The findings of this study provide an opportunity to improve understanding of the morphological and genetic diversity of ciliates.

## 1. Introduction

Ciliates are single-celled eukaryotes that are highly developed, ubiquitous in freshwater, marine and terrestrial as biotopes, speciose, and morphologically diverse [1,2,3,4,5]. Since Antonie van Leeuwenhoek made the first descriptions of ciliates, approximately 8000 free-living and epibiotic species have been recognized [6]. Most of them are free-living species, and others are symbionts, either commensals or parasites, mostly on the external surfaces of their hosts but sometimes internally [7]. In the first 200 years, the taxonomic identifications or descriptions of ciliates were mainly based on living morphology. However, their morphological features can sometimes reflect certain aspects of prevailing environmental conditions, such as food abundance and the presence of predators [8,9,10,11]. Consequently, populations of the same species may present high morphological plasticity, resulting in significant problems in taxonomic identification [5,9,10,12,13,14,15]. Over the last century, introduction of silver staining methods provided recognition of ciliary patterns and nuclear morphology; these detailed structures allowed a better taxonomy of ciliates [6,16]. In recent years, molecular technologies have been widely used in ciliate research and have greatly facilitated the study of ciliate taxonomy and systematics. However, there are numerous examples in ciliate research showing that morphological and molecular evolutions are not always concordant [10,17]. In many cases, taxa with similar or even identical morphology show highly divergent molecular information, while diverse morphotypes share the same molecular information. These anomalies have brought new controversies to ciliate research [12,17].

Cyrtophorians are highly divergent ciliates with dorsoventrally or laterally flattened bodies. Since the beginning of the 21st century, numerous cyrtophorians have been reported from various habitats, and many of these have been used as model organisms in morphological and genetic diversity research [1,5,10,12,18,19,20,21,22]. *Dysteria* Huxley, 1857, is a specialized cyrtophorian genus with a highly bilaterally compressed body and somatic cilia densely arranged in kineties that are restricted to a narrow ventral groove between two lateral plates. Other diagnostic features include the right body plate, which is arched and is slightly larger than the flattened left plate and the podite, which is located at the rear end of the sulcus and is an attachment organelle that enables an organism to attach to submerged substrata, such as rocks and plants [2,5,10,19,23,24,25,26]. Relatively few morphological features of *Dysteria* (e.g., body shape, cell size, dorsal spine) can be used for species identification, which has led to many misidentifications, synonyms, and the recognition of cryptic species [5,9,10,12,24,25,26,27,28,29]. In the last two decades, the application of silver staining methods and molecular technology has enabled several historical confusions and errors to be resolved and for new species to be described [10,12,30]. To date, about 45 nominal *Dysteria* species (including *Dysteria paracrassipes* n. sp.) have been recorded, 26 of which have been investigated using staining methods including the type species *D. armata* [31]. The species and genetic diversity of *Dysteria*, therefore, remain poorly understood.

In this study, three *Dysteria* species (i.e., *D. paracrassipes* n. sp., *D. crassipes,* and *D. brasiliensis*) were isolated from subtropical brackish wetlands in China. Their morphological features were investigated based on observations in vivo and following protargol staining. The small subunit ribosomal DNA (SSU rDNA) of each was sequenced, and the molecular phylogeny of the genus *Dysteria* was analyzed. The main aim of the study is to improve knowledge and understanding of the morphological and molecular diversity of *Dysteria*.

## 2. Results

Class Phyllopharyngea de Puytorac et al., 1974

Subclass Cyrtophoria Fauré-Fremiet in Corliss, 1956

Order Dysteriida Deroux, 1976

Family Dysteriidae Claparède & Lachmann, 1858

Genus *Dysteria* Huxley, 1857

### 2.1. Dysteria paracrassipes n. sp.

#### 2.1.1. ZooBank Registration

Present work: urn:lsid:zoobank.org:pub:DD76F78C-4EB7-49F6-81DD-F855B8A65101

New species: urn:lsid:zoobank.org:act:48A6961F-11E3-4179-9712-44B37A2735F1

#### 2.1.2. Diagnosis

Body 45–60 × 30–35 μm in vivo, oval in outline with both ends rounded; four right kineties including two frontoventral kineties, innermost row commencing below cytostome and terminating at level of podite; four to eight left kineties, densely arranged near equatorial area; two parallel circumoral kineties almost equal in length; one double-rowed preoral kinety, kinetosomes obliquely arranged in front of cytostome; three left frontal kineties between preoral kinety and circumoral kineties; brackish water habitat.

#### 2.1.3. Type Locality

A subtropical coastal wetland in Ningbo, China, East China Sea (29°46′23″ N, 121°57′17″ E), where the water temperature was about 26 °C and salinity was 4.5.

#### 2.1.4. Etymology

The species-group name “*paracrassipes*” is a composite of the Greek adjective “*para*-” (beside) and the species-group name “*crassipes*”, indicating its similarity to *Dysteria crassipes.*

#### 2.1.5. Type Materials

Seven slides with protargol-stained specimens have been deposited in the Laboratory of Protozoology, Ocean University of China (OUC), including one slide (registration number: YTT-20200528-01) with the holotype specimen circled in black ink and six slides (registration numbers: YTT-20200528-02, 03…07) with paratype specimens.

#### 2.1.6. Morphological Description

Body 45–60 × 30–35 μm in vivo, bilaterally compressed, oval in outline when viewed from lateral aspect, both ends broadly rounded, ventral side straight and dorsal side slightly convex (Figure 1A and Figure 2A). When observed from ventral aspect, right body plate arched and slightly larger than flattened left plate (Figure 1D and Figure 2B). Fine equatorial transverse stripe located at midbody on left plate (Figure 2C). Cytoplasm colorless to grayish, filled with different-sized food vacuoles, lipid droplets, and ingested diatoms (Figure 1A and Figure 2A). Cytostome ventrally located in anterior 1/5 of body. Two pharyngeal rods, each about 20 μm in length, extending to about posterior quarter of body. Two contractile vacuoles, each about 5–7 µm in diameter, ventrally positioned, one in anterior third, the other in posterior third of body; each pulses at an interval of 1–2 min (Figure 1A,C and Figure 2A). Single ovoidal macronucleus, 12–27 × 12–19 μm after protargol staining, positioned in midregion of body, characteristically heteromeric (Figure 1A,C,E and Figure 2D). Micronucleus not detected. Podite slender, about 8 μm long in vivo, located at rear end of ventral sulcus. Dorsal spine not detected. Locomotion usually by crawling on substrate.

Ciliature as shown in Figure 1B,E and Figure 2D–G. Four right kineties including two frontoventral kineties and two inner kineties, progressively shorter in length from right to left; longest frontoventral kineties extend anteriorly to dorsal margin, each composed of 84–110 basal bodies; two inner kineties commence near cytostome, innermost row terminating at level of podite and composed of 46–55 basal bodies (Figure 1E and Figure 2D, Table 1). Cilia of right kineties about 12 μm long in vivo. Four to eight left kineties, densely arranged near equatorial area (Figure 1E and Figure 2G). Equatorial fragment composed of 4–18 basal bodies (Figure 1E and Figure 2D,E, Table 1), cilia about 9 μm long in vivo. Terminal fragment positioned near front end of frontoventral kineties and composed of 4–7 basal bodies (Figure 1B and Figure 2F, Table 1). Two single-rowed circumoral kineties nearly equal in length and transversely arranged in parallel, located near cytostome (Figure 1B and Figure 2E). Double-rowed preoral kinety, obliquely arranged in front of cytostome (Figure 1B and Figure 2E). Three single-rowed left frontal kineties, transversely oriented, between preoral kinety and circumoral kineties (Figure 1B,E and Figure 2E). Glandule visible in protargol-stained specimens, about 3–6 μm in diameter, located near rear end of innermost right kinety (Figure 1E, Table 1).

### 2.2. Dysteria Crassipes Claparède & Lachmann, 1859

*Dysteria crassipes* was briefly described in an original report by Claparède and Lachmann [32]. Kahl [21] provided some supplementary morphological information in his short redescription. Gong et al. [9] supplied the first detailed description, including both the living morphology and ciliature, and an improved diagnosis. In recent years, several temperate and tropical populations with similar morphological characteristics to those described previously have been reported [5,12]. Here, we supply a detailed description of the morphology of a subtropical population.

#### 2.2.1. Voucher Slides

Six voucher slides (registration numbers: JDD-20190226-3-01, 02…06) with protargol-stained specimens have been deposited in the Laboratory of Protozoology, Ocean University of China (OUC).

#### 2.2.2. Morphological Description of Ningbo Population

Body 50–65 × 35–45 μm in vivo, bilaterally compressed, approximately rectangular in outline with both ends rounded, right plate slightly larger than left plate (Figure 1F and Figure 2H). Ventral margin straight, dorsal margin slightly convex near equator. Thin, transversely oriented stripe near equator on left plate (Figure 2J). Cytoplasm colorless to grayish, filled with food vacuoles and lipid droplets (Figure 1F and Figure 2H). Cytostome about 5 µm in diameter, located about 25% down length of body. Two pharyngeal rods, conspicuous in vivo, each about 20 µm in length (Figure 2I). Two contractile vacuoles each about 6–8 µm in diameter, usually one in anterior third and the other in posterior third of body (Figure 1F,H and Figure 2H). Podite about 8–10 μm long in vivo, positioned in posterior 1/5 of body (Figure 2K). Single heteromeric macronucleus located in midbody region. Micronucleus not observed. Locomotion by slowly crawling on substrate.

Ciliature, as shown in Figure 1G,K and Figure 2L–N. Four right kineties, including two frontoventral kineties and two inner kineties that are progressively shorter from right to left; innermost right kinety commences about 25% down length of body and terminates near base of podite (Figure 1K and Figure 2L). Two frontoventral kineties almost equal in length and extend anteriorly to dorsal margin, each composed of 93–131 basal bodies (Figure 1K and Figure 2L). Five to nine left kineties, densely arranged in equatorial area (Figure 1K and Figure 2N). Equatorial fragment composed of 5–20 basal bodies (Figure 1K and Figure 2L,N). Terminal fragment short and curved, containing 3–7 basal bodies (Figure 1G and Figure 2L). Two circumoral kineties nearly equal in length and transversely oriented in parallel. Single-rowed preoral kinety, obliquely arranged in front of cytostome. Three left frontal kineties transversely oriented between preoral kinety and circumoral kineties (Figure 1G and Figure 2M). Glandule conspicuous after protargol staining, about 3–6 μm in diameter, located near rear end of innermost right kinety (Figure 1K and Figure 2L).

### 2.3. Dysteria Brasiliensis Faria et al., 1922

*Dysteria brasiliensis* has been redescribed many times since it was originally reported by Faria et al. [33]. However, the morphological characteristics of the various populations differ. Here, we supply details of the morphology of five populations collected from subtropical coastal waters in China.

#### 2.3.1. Voucher Slides

Voucher slides with protargol-stained specimens are deposited in the Laboratory of Protozoology, Ocean University of China (OUC), with registration numbers as follows: population-I (JDD-20190521-3-01, 02…07), population-II (YTT-20190723-3-01, 02…05), population-III (ZXT-20200608-2-01, 02…06), population-IV (ZXT-20200612-1-01, 02…07), population-V (ZXT-20200720-1-01, 02…06).

#### 2.3.2. Morphological Description of Ningbo Population-I 

This population is mainly illustrated in Figure 3F,G and Figure 4D,G,I,M–O. Body about 105–125 × 40–55 μm in vivo, bilaterally compressed, when viewed from lateral aspect triangular to elongate rectangular, front end blunt and round, posterior end slightly narrowed, ratio of width to length about 1:3, ventral margin straight, dorsal margin slightly convex at the equator, right plate slightly larger than left plate (Figure 3F and Figure 4D). Dorsal spine conspicuous, hook-shaped, about 10–15 μm long, located at posterior end of cell (Figure 4I). Cytoplasm colorless, filled with numerous food vacuoles and a few lipid droplets. Cytostome conspicuous in vivo, located about 20% down length of body; two strong and conspicuous pharyngeal rods extending obliquely to posterior end of body, each about 35 μm long (Figure 3F). Two contractile vacuoles, each about 5 μm in diameter, one located in anterior quarter and the other in posterior quarter of cell. Podite slender, 8–10 μm long in vivo, located at rear end of ventral sulcus (Figure 4H). Macronucleus ellipsoidal and heteromeric, micronucleus not detected (Figure 3F and Figure 4D). Both plates densely covered by rod-shaped bacteria (Figure 4G). 

Ciliature as shown in Figure 3G and Figure 4I,M–O. Five right kineties, including two frontoventral kineties (Figure 3G). Two frontoventral kineties, equal in length, extending almost entire body length, each containing about 182–293 basal bodies, cilia about 10 μm in vivo. The other three right kineties commence near cytostome but terminate at different levels, becoming progressively shorter from right to left, innermost kinety being shortest (Figure 3G). Five to nine left kineties closely arranged in midregion of body. Equatorial fragment composed of 7–33 kinetosomes (Figure 4M). Terminal fragment arc-shaped, consisting of 9–17 basal bodies. Two circumoral kineties almost equal in length and arranged in parallel, obliquely oriented above cytostome. Single preoral kinety, obliquely oriented. Three horizontal left frontal kineties orthogonal to and optically intersect the two circumoral kineties (Figure 3G).

#### 2.3.3. Morphological Description of Ningbo Population-II

This population (Figure 4A) closely resembles Ningbo population-I except the body size in vivo (115–135 × 45–55 μm vs. 105–125 × 40–55 μm) and the number of basal bodies in each frontoventral kinety (206–293, mean about 249 vs. 182–283, mean about 232). In addition, this population has no dorsal spine.

#### 2.3.4. Morphological Description of Ningbo Population-III 

This population (Figure 3D,E and Figure 4E,F,K) is characterized by the dominant dorsal spine, which is 35–40 μm long in vivo. The ratio of dorsal spine length to body length is about 30%. This population differs from Ningbo population-I in other morphological characters as follows: (1) larger body size in vivo (170–210 × 40–50 μm vs. 105–125 × 35–50 μm); (2) more slender body, as the ratio of width to length is about 1:4 (vs. 1:3); and (3) more basal bodies in the frontoventral kinety (188–353, mean about 270, vs. 182–283, mean about 232).

#### 2.3.5. Morphological Description of Ningbo Population-IV

The main feature of this population (Figure 3A–C and Figure 4B,J) is the hockey-stick-shaped body; that is, the truncated anterior end bends from ventral to dorsal, forming a prominent right angle. The equatorial region of the body is not conspicuously bulged, but gradually tapers from the front to the posterior end. The slender posterior end lacks a dorsal spine. In a small number of individuals (5 out of 16), there are six right kineties, whereas in other populations there are invariably five right kineties.

#### 2.3.6. Morphological Description of Ningbo Population-V

This population (Figure 4C,H) agrees well with Ningbo population-I except the slightly larger body size in vivo (120–160 × 35–55 μm vs. 105–125 × 35–50 μm) and the shorter dorsal spine (8–11 μm vs. 9–20 μm).

### 2.4. SSU rRNA Gene Sequences and Phylogenetic Analyses

The SSU rDNA sequences derived from the three *Dysteria* species were submitted to the NCBI database with lengths, GC contents, and GenBank accession numbers as follows: *Dysteria paracrassipes* n. sp. (1523 bp, 45.04%, OL527698), *D. crassipes* (1560 bp, 45.07%, OL527699). *D. brasiliensis*: Ningbo population-I (1564 bp, 44.57%, OL527704), Ningbo population-II (1564 bp, 44.63%, OL527703), Ningbo population-III (1563 bp, 44.59%, OL527700), Ningbo population-IV (1648 bp, 44.17%, OL527701), Ningbo population-V (1594 bp, 44.60%, OL527702). A BlastN analysis of the new sequences against the NCBI database showed that the sequence of *D. paracrassipes* n. sp. is most similar to those of *Dysteria* sp. KY922819 (99.03%) and *D. cristata* KC753488 (98.97%), and the sequence of *D. crassipes* was most similar to those of *D. crassipes* KC753493 (98.93%) and *D. crassipes* KC753492 (98.67%). 

Comparing the SSU rDNA sequences of these populations, the Ningbo population of *D. crassipes* differed by 18–25 bases from its congener populations (corresponding to 98.93%–98.48% similarity). In contrast, the SSU rDNA sequences among the nine populations of *D. brasiliensis* (five of which were derived from the present study) were more conserved with a maximum difference of only 6 bases (Figure 5).

The maximum likelihood (ML) and Bayesian inference (BI) trees had nearly identical topologies; therefore, only the ML tree is shown here with nodal support values from both algorithms (Figure 6). In the SSU rDNA tree, the new sequence of *D. paracrassipes* n. sp. clusters with *D. cristata* (KC753488) and *Dysteria* sp. (KY922819) with full support (ML/BI, 100%/1.00). The new sequence of *D. crassipes* (Ningbo population) is sister to a fully supported clade of *D. crassipes* populations (ML/BI, 97%/1.00). The five newly sequenced *D. brasiliensis* populations, three known *D. brasiliensis* populations, and another four unidentified *Dysteria* species form a strongly supported clade (ML/BI, 90%/0.96) that is sister to the *D. compressa* + *D. monostyla* clade.

## 3. Discussion

*Dysteria* is one of the most commonly reported cyrtophorian genera, and due to continuous studies of its morphology and phylogeny, especially during the last two decades, a high species diversity of this genus has been revealed [9,23,24,25,27,34]. In addition to the establishment of new species, many poorly known species have been redescribed using modern methods, including live observation, silver staining, and molecular analysis [5,10]. This in turn brings challenges to species circumscription and identification when there is a discrepancy between the morphological and molecular data. Research on different populations of the same nominal morphospecies has revealed examples both of intra-specific variations and cryptospecies among morphologically similar but molecularly different populations [10]. The present work describes one new species and several populations of two known species. By making detailed comparisons among species and populations using both morphological and molecular data, we hope to improve understanding of the species diversity in *Dysteria*.

### 3.1. Comments on Dysteria paracrassipes n. sp.

With reference to its living morphological characteristics and ciliature, *Dysteria paracrassipes* n. sp. closely resembles *D. crassipes* Claparède & Lachmann, 1859. However, there is a significant difference between these two species; that is, the preoral kinety is double-rowed in *D. paracrassipes* n. sp. (vs. single-rowed in *D. crassipes*). In previous studies, the structure and position of the preoral kinety were largely ignored. We carefully examined and verified this structure of many well illustrated *D. crassipes* populations, all of which are single-rowed [9]. Therefore, this is a stable and reliable character for separating these two species. In addition, the SSU rDNA sequence of *D. paracrassipes* n. sp. differs significantly and shares a poor genetic similarity with the SSU rDNA sequences of the *D. crassipes* populations (Ningbo population, 56 bases, 96.38%; Zhuhai population, 56 bases, 96.20%; Haikou population, 55 bases, 96.47%; Shenzhen population, 55 bases, 96.33%; Daya Bay population, 57 bases, 96.46%), providing further support for the validity of the new species.

### 3.2. Comments on Dysteria crassipes Claparède & Lachmann, 1859

*Dysteria**crassipes* was originally reported by Claparède and Lachmann [32] and briefly redescribed by Kahl [21]. Its morphology, however, remained poorly known with information confined to features such as body size and shape. Gong et al. [9] provided the first detailed morphological description using modern methods based on a population collected from Chinese coastal waters of the Yellow Sea. *Dysteria*
*crassipes* has since been reported repeatedly in coastal waters of China, for example, by Chen et al. [17], who documented the SSU rDNA sequences of three populations from the South China Sea but without morphological data, and Wang et al. [5], who provided morphological and molecular information of a South China Sea population.

The Ningbo population collected from subtropical waters of East China Sea is morphologically consistent both with the Qingdao populations (temperate coastal waters of the Yellow Sea) and the Haikou populations (tropical waters of the South China Sea) [5,9,10]. However, the SSU rDNA sequences of these Chinese populations are quite different; that is, the number of base-pair differences ranges from 5 to 25, and sequence identities among the populations range from 98.39% to 99.71% (Appendix A, Figure 5). Because of the absence of distinct species-level morphological differences, we consider the current isolate to be a subtropical population of *D. crassipes.* However, it seems likely that there are several cryptic species within the *D. crassipes* complex.

### 3.3. Comments on Dysteria Brasiliensis Faria et al., 1922

*Dysteria brasiliensis* is a well-known species that has been investigated many times since being originally described by Faria et al. [33]. Song and Packroff [34] provided the first detailed morphological description using modern methods. In their investigation of different Yellow Sea populations, Gong et al. [9] showed that the morphology of this species is highly variable, for example, in terms of the body shape and the presence/absence of the dorsal spine. Qu et al. [10] added some morphological information based on two populations, one collected from the Bohai Sea and the other from the South China Sea [10]. Wang et al. [12] provided morphological and molecular information based on another tropical population, further enriching knowledge of the morphological and molecular diversity of this species.

The present work includes a comprehensive study on the morphological and molecular data of five subtropical populations of *D. brasiliensis*. A comparison of the four known and five new SSU rDNA sequences revealed that the molecular information is highly consistent, with a maximum difference of only six bases and sequence identities of 99.62% to 100% (Figure 5). However, there were significant morphological variations among these populations, mainly in body shape and dorsal spine structure (Appendix A). These could be roughly divided into three types (i.e., slim and curved (Figure 3A,K,L), slim with a prominent dorsal spine (Figure 3D,M,N), and triangular or ellipsoidal with a curved dorsal spine (Figure 3F,H,J). Similarly, the characteristics of the dorsal spine also vary greatly. Among the five populations collected from Ningbo, some individuals had no dorsal spine, some had a short hook-shaped dorsal spine, and some had a long spear-shaped dorsal spine that was up to one-third of the body length (Figure 3D,E,N and Figure 4E,F). Some researchers have regarded the dorsal spine structure as a species character [21,32], whereas Gong et al. [9] did not accept this assertion, noting that the dorsal spine was not a key feature for *Dysteria* identification. Based on the morphological and molecular information of multiple populations of *D. brasiliensis*, we suggest that variations in the body shape and features of the dorsal spine of *Dysteria* are probably responses to the environment, including food abundance and predator pressure.

### 3.4. Phylogenetic Analyses Based on SSU rDNA Sequences

As shown in Figure 6, *Dysteria paracrassipes* n. sp. is sister to the *D. cristata* KC753488 + *Dysteria* sp. KY922819 clade. Furthermore, most of the morphological features of the new species are similar with *Dysteria cristata* (e.g., ovoidal body shape, approximately 45–60 μm × 30–35 μm in vivo, with four right kineties and four to eight left kineties), thus supporting the phylogenetic affiliation of these two taxa. However, *D. paracrassipes* differs from *D. cristata* in having four right kineties (vs. three in *D. cristata*) [10,27]. Although the five *D. brasiliensis* populations investigated here differ slightly in the base differences of their SSU rDNA (maximum difference of six bases) and cluster together in the phylogenetic tree, they vary greatly in their morphology (i.e., significant differences in body size and shape, dorsal spine, and ratio of dorsal spine to body length (Appendix A)). Accordingly, we speculate that the presence and shape of the dorsal spine of *D. brasiliensis* do not correspond to systematic signals in the SSU rDNA sequences and are not taxonomically informative characters.

## 4. Materials and Methods

### 4.1. Sample Collection, Observation, and Identification

Three *Dysteria* species were collected on 28 May 2020 from subtropical brackish coastal waters of the East China Sea at Ningbo, China (Figure 7). *Dysteria paracrassipes* n. sp. was collected from a coastal wetland on Meishan Island (29°46′23″ N, 121°57′17″ E), where the water temperature was about 26 °C and salinity was 4.5. *Dysteria crassipes* was collected on 26 February 2019 from a brackish lake (29°45′51″ N, 121°54′2″ E) connected to the East China Sea by channels, where the water temperature was about 12 °C and salinity was 20.0. Five populations of *D. brasiliensis* were collected from 2019 to 2020: population-I was collected from a brackish fish-culturing pond (29°33′50″ N, 121°42′49″ E), where the water temperature was about 28 °C and salinity was 26.0; populations II–V were collected from the same brackish lake as above (29°45′51″ N, 121°54′2″ E) in different seasons, so the water temperature varied between 28 and 34 °C and the salinity ranged from 17.0 to 19.0. Samples were taken from the surface of the sediment using a sterile syringe and the dilute with untreated water from the collection site. Clonal cultures were maintained for a few days in Petri dishes at room temperature using filtered habitat water. Rice grains or wheat grass juice was added to promote the growth of bacteria as food source for the ciliates. All cultured ciliated died within about 1 week; therefore, we were not able to maintain either taxon for a long time. However, it was possible to isolate enough individuals of three species to provide a detailed morphological description.

Cells were observed in vivo using bright field and differential interference contrast (DIC) microscopy (Leica DM2500) at 400–1000× magnification. Protargol staining was used to reveal the ciliature and nuclear apparatus following the method of Wilbert [35]. Counts, measurements, and drawings of stained specimens were performed at 1000× magnification.

### 4.2. DNA Extraction, PCR Amplification, and Gene Sequencing

For each population, a single cell was isolated using a micropipette. Specimens were washed three to five times using filtered habitat water and twice using ultrapure water. Three to five parallel molecular samples of every population would be prepared. Extraction of genomic DNA was performed using a DNeasy Blood and Tissue Kit (Qiagen, Hilden, Germany), following the manufacturer’s instructions. Two samples were randomly chosen and sent for sequencing. No intraspecific variations were identified. The SSU rDNA was amplified using the primers 18SF (5′-AAC CTG GTT GAT CCT GCCi AGT-3′) and 18SR (5′-TGA TCC TTC TGC AGG TTC ACC TAC-3′) [36]. To minimize the possibility of amplification errors, a Q5 Hot Start High-Fidelity DNA Polymerase (New England Biolabs Co., Ltd., M0493, Beijing) was used.

The polymerase chain reaction (PCR) conditions used in the amplification were as follows: initial denaturation for 30 s at 98 °C, 35 cycles of 10 s, denaturation for 20 s at 98 °C, primer annealing for 100 s at 56 °C, primer elongation at 72 °C, final primer elongation for 5 min at 72 °C. Sequencing was performed bidirectionally at Tsingke Biological Technology Company (Beijing, China).

### 4.3. Phylogenetic Analyses

A total of 51 SSU rDNA sequences were selected for the phylogenetic analyses, including the 7 newly obtained sequences and 44 reference sequences downloaded from the National Center for Biotechnology Information (NCBI) database (for accession numbers, see Figure 6). Six chlamydodontids, namely, *Chlamydodon triquetrus* (MG566058), *C. similis* (KY496621), *C. oligochaetus* (KY496620), *C. rectus* (KT461932), *C. caudatus* (JQ904058), and *C. wilberti* (MG566060), were assigned as the out-group. The 51 sequences were aligned using MUSCLE [37] with default parameter settings and were then manually edited using the program BioEdit 7.0.5.3 [38]. Maximum likelihood (ML) analysis was conducted on the CIPRES Science Gateway server using RAxML-HPC2 located on XSEDE v8.2.9 [39], with the GTR + I + G model. Support for the best ML tree was calculated from 1000 bootstrap replicates. Bayesian inference (BI) analysis was performed using MrBayes on XSEDE v3.2.6 [40]. The GTR + I + G model selected by MrModeltest v2.2 was applied in the BI analysis [41]. The BI analysis was run for 10^6^ generations with trees sampled every 100th generation, with the first 2500 trees discarded as burn-in. The tree topologies were visualized via MEGA v7.0 and TreeView v.1.6.6 [42,43]. Systematic classification and terminology mainly followed Lynn [6], Gao et al. [44], and Wang et al. [12].

## Figures and Tables

**Figure 1 ijms-23-01764-f001:**
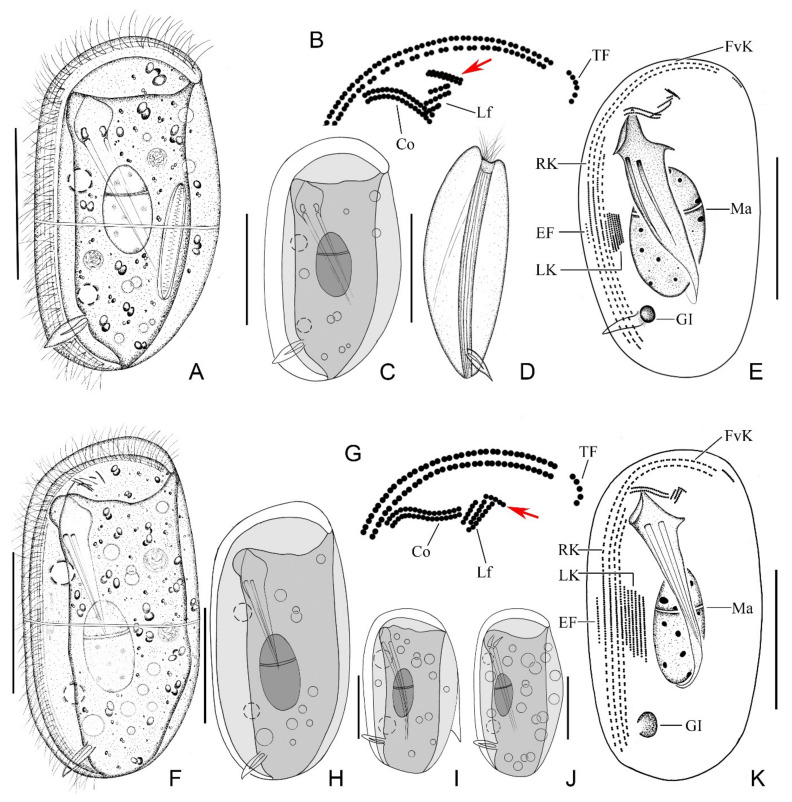
*Dysteria paracrassipes* n. sp. from life (**A**,**C**,**D**) and after protargol staining (**B**,**E**). *Dysteria crassipes* from life (**F**,**H**–**J**) and after protargol staining (**G**,**K**). (**A**) Left lateral view of a representative individual. (**B**) Details of anterior region of cell showing the ciliary pattern; arrow shows double-rowed preoral kinety. (**C**) Showing the body shape. (**D**) Ventral view. (**E**) Left lateral view of the holotype specimen. (**F**) Left lateral view of a representative individual. (**G**) Details of anterior region of cell showing the ciliary pattern; arrow shows single-rowed preoral kinety. (**H**) Left lateral view of typical individual. (**I**,**J**) Representative individuals after Gong et al. [9]; note that the individual in (**I**) has a dorsal spine. (**K**) Left view showing the infraciliature. Co, circumoral kineties; EF, equatorial fragment; FvK, frontoventral kineties; GI, glandule; Lf, left front kineties; LK, left kineties. Ma, macronucleus; P, podite; Pr, preoral kinety; RK, right kineties; TF, terminal fragment. Scale bars = 25 μm.

**Figure 2 ijms-23-01764-f002:**
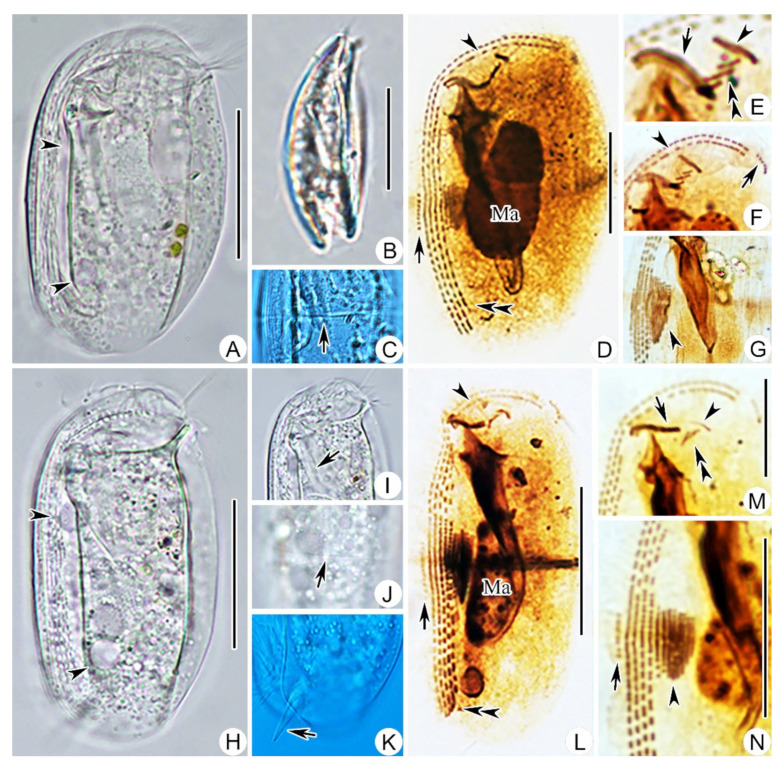
Photomicrographs of *Dysteria paracrassipes* n. sp. (**A**–**G**) and *Dysteria crassipes* (**H**–**N**) from life (**A**–**C**,**H**–**K**) and after protargol staining (**D**–**G**,**L**–**N**). (**A**) Left lateral view of a representative individual; arrowheads point to contractile vacuoles. (**B**) Ventral view. (**C**) To show the equatorial transverse stripe (arrowhead). (**D**) Left lateral view of the holotype specimen; arrow shows equatorial fragment; arrowheads mark frontoventral kineties and double arrowhead points to right kineties. (**E**) Oral ciliary pattern; arrow shows circumoral kineties; arrowheads mark double-rowed preoral kinety and double arrowhead points to left front kineties. (**F**) Anterior end of cell; arrow shows equatorial fragment; arrowheads mark frontoventral kineties. (**G**) Ciliary pattern of midbody; arrowhead marks right kineties. (**H**) Left lateral view of a representative individual; arrowheads point to contractile vacuoles. (**I**) Focusing on surface of left side; arrow shows one of the cytopharyngeal rods. (**J**) To show the equatorial transverse stripe (arrowhead). (**K**) Posterior portion of cell; arrow shows podite. (**L**) General view of infraciliature; arrow shows equatorial fragment; arrowhead marks frontoventral kineties and double arrowhead points to right kineties. (**M**) Oral ciliary pattern; arrow shows circumoral kineties; arrowhead marks single-rowed preoral kinety and double arrowhead points to left front kineties. (**N**) Ciliary pattern of midbody; arrowhead marks right kineties; arrow shows equatorial fragment. Ma, macronucleus. Scale bar = 25 μm.

**Figure 3 ijms-23-01764-f003:**
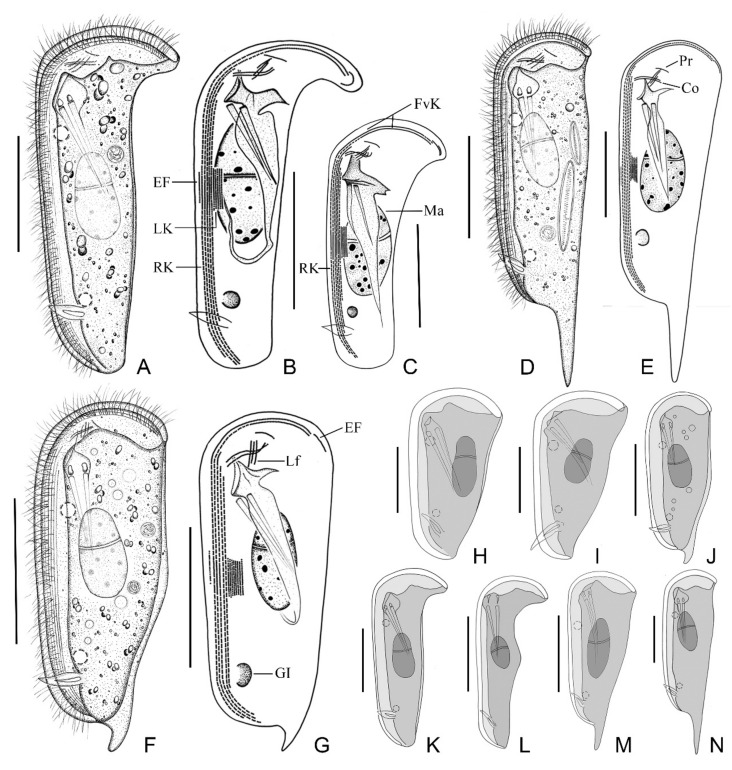
*Dysteria brasiliensis* from life (**A**,**D**,**F**,**H**–**N**) and after protargol staining (**B**,**C**,**E**,**F**). (**A**) Left lateral view; note that anterior portion is distinctly wider than other parts of cell. (**B**,**C**) Left view showing infraciliature; note that one cell has five right kineties (**B**) and the other has six right kineties (**C**). (**D**) Left lateral view of an individual with a long dorsal spine. (**E**) Left view showing infraciliature of an individual with a long dorsal spine. (**F**) Left lateral view of a representative individual. (**G**) Left view showing infraciliature. (**H**–**N**) Left lateral views of differently shaped individuals (**H**,**I**,**L**,**M**) from Gong et al. [9]. Co, circumoral kineties; EF, equatorial fragment; FvK, frontoventral kineties; GI, glandule; Lf, left front kineties; LK, left kineties; Ma, macronucleus; P, podite; Pr, preoral kinety; RK, right kineties; TF, terminal fragment. Scale bars = 80 μm (**A**–**G**); 30 μm (**H**,**I**); 60 μm (**J**–**N**).

**Figure 4 ijms-23-01764-f004:**
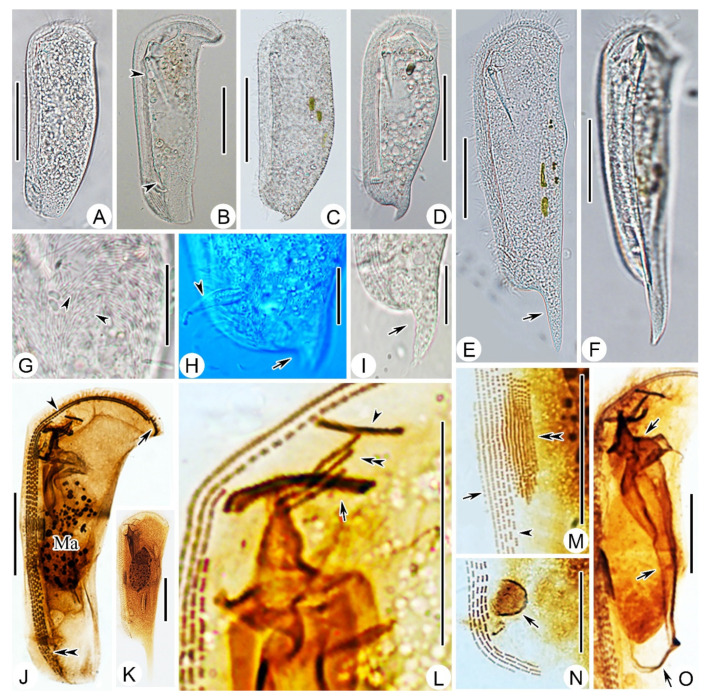
Photomicrographs of *Dysteria brasiliensis* in vivo (Ningbo population-II (**A**), Ningbo population-III (**B**), Ningbo population-V (**C**,**H**), Ningbo population-I (**D**,**G**), Ningbo population-IV (**E**,**F**), and after protargol staining (**J**–**O**). (**A**) Left lateral view of representative individual. (**B**) Left lateral view, to show the absence of a dorsal spine, and that anterior portion is distinctly wider than other parts of cell; arrowheads point to contractile vacuoles. (**C**–**E**) Left lateral view, to show the variable cell outline and with different-length dorsal spine (arrow). (**F**) Ventral view. (**G**) Arrowheads indicate rod-shaped bacteria on cell surface. (**H**,**I**) Left lateral view showing different-length dorsal spine (arrows) and podite (arrowhead). (**J**,**K**) Lateral views of two different specimens showing infraciliature; arrow shows equatorial fragment; arrowhead marks frontoventral kineties; double arrowhead points to right kineties. (**L**) Oral ciliary pattern; arrow shows circumoral kineties; arrowhead marks preoral kinety; double arrowhead points to left front kineties. (**M**) Ciliary pattern of midbody region; arrow shows equatorial fragment; arrowhead marks right kineties; double arrowhead points to left kineties. (**N**) Posterior end of ventral groove; arrow shows glandule. (**O**) Lateral view of oral region; arrow shows the winding of the cyrtos. Ma, macronucleus. Scale bar = 50 μm (**A**–**F**,**G**,**K**); 20 μm (**G**–**I**,**L**–**O**).

**Figure 5 ijms-23-01764-f005:**
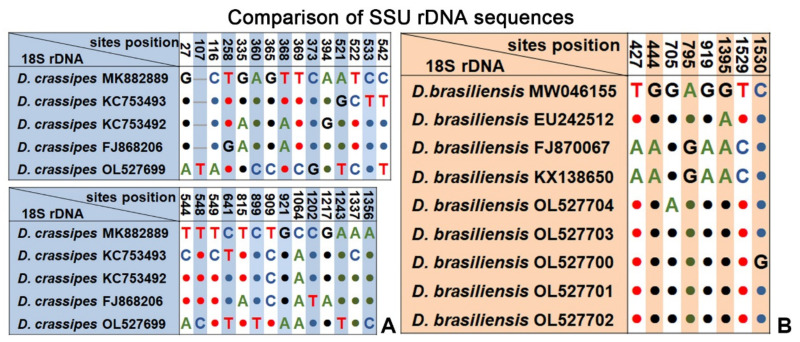
Sequence comparison of the SSU rRNA gene showing the unmatched nucleotides between the Ningbo population of *Dysteria crassipes* and other populations of *D. crassipes* (**A**) and the unmatched nucleotides among the Ningbo populations of *D. brasiliensis* and other populations of *D. brasiliensis* (**B**). Nucleotide positions are given at the top of each column. Insertions and deletions are compensated by introducing alignment gaps (-). Matched sites are represented by dots (·).

**Figure 6 ijms-23-01764-f006:**
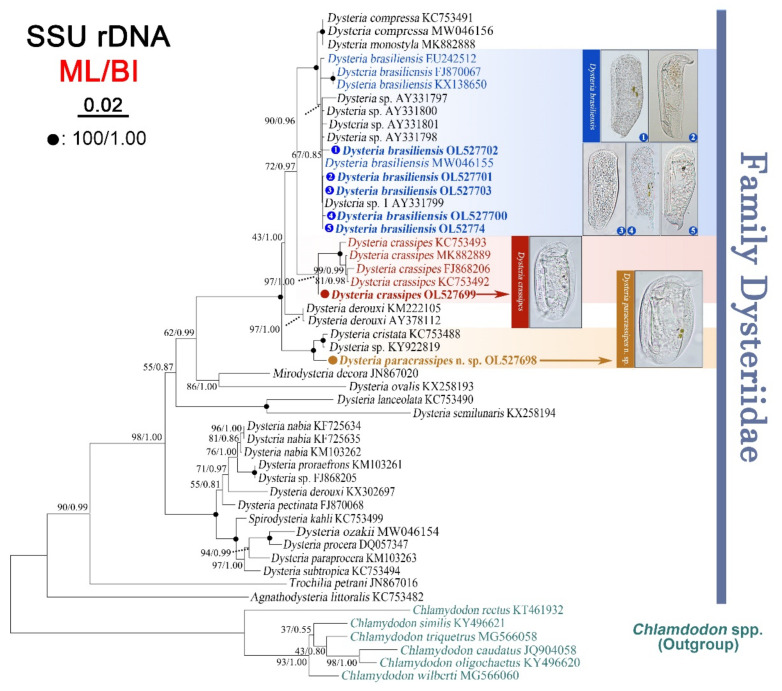
Maximum likelihood (ML) tree inferred from SSU rDNA sequences showing positions of *Dysteria paracrassipes* n. sp., *Dysteria crassipes*, and *Dysteria brasiliensis* (bold font). Numbers at nodes indicate the bootstrap values of maximum likelihood (ML) out of 1000 replicates and the posterior probabilities of Bayesian analysis (BI). Solid circles represent full bootstrap supports from both algorithms. The scale bar corresponds to two substitutions per 100 nucleotide positions.

**Figure 7 ijms-23-01764-f007:**
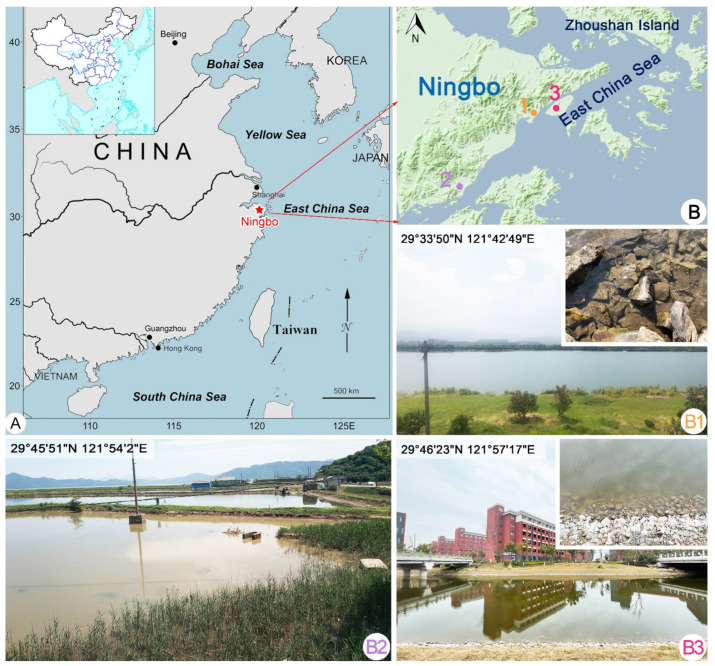
Maps showing the location of Ningbo and the sampling locations in Ningbo and photographs of the sampling sites. (**A**) Map of China showing the location of Ningbo. (**B**) Portion of the map of China showing the location of sampling sites in Ningbo (colored circles and numbers). (**B1**) Sampling site for *Dysteria brasiliensis* (Ningbo populations II–V) and *Dysteria crassipes*. (**B2**) Sampling site for *Dysteria brasiliensis* (Ningbo population-I). (**B3**) Sampling site for *Dysteria paracrassipes* n. sp.

**Table 1 ijms-23-01764-t001:** Morphometric data on *Dysteria paracrassipes* n. sp. (par), *D. crassipes* (cra, Ningbo population), and *D. brasiliensis* (bra1, Ningbo population-I; bra2, Ningbo population-II; bra3, Ningbo population-III; bra4, Ningbo population-IV; bra5, Ningbo population-V).

Characters	Species	Max	Min	Mean	M	SD	CV%	*n*
Body length (μm)	bra1	133	87	110.9	111	2.52	2.27	25
	bra2	144	104	121.2	118	9.82	8.10	25
	bra3	145	101	118.2	119	10.86	9.19	25
	bra4	240	177	204.1	206	16.39	8.03	16
	bra5	157	96	126.5	126	13.49	10.66	25
	cra	68	45	57.8	57	1.00	1.73	25
	par	56	42	50.6	51	3.91	7.73	25
Body width (μm)	bra1	49	30	39.3	40	0.94	2.39	25
	bra2	58	44	50.3	50	3.20	6.36	25
	bra3	58	42	49.8	49	3.75	7.53	25
	bra4	64	43	52.4	51.5	6.12	11.68	16
	bra5	63	42	51.4	52	5.51	10.72	25
	cra	68	27	33.7	32	1.49	4.41	25
	par	36	24	31.2	32	2.79	8.94	25
Number of right kineties	bra1	5	5	5.0	5	0.00	0.00	25
	bra2	5	5	5.0	5	0.00	0.00	25
	bra3	6	5	5.2	5	0.41	7.88	25
	bra4	5	5	5.0	5	0.00	0.00	16
	bra5	5	5	5.0	5	0.00	0.00	25
	cra	4	4	4.0	4	0.00	0.00	25
	par	4	4	4.0	4	0.00	0.00	25
Number of frontoventral kineties	bra1	2	2	2.0	2	0.00	0.00	25
	bra2	2	2	2.0	2	0.00	0.00	25
	bra3	2	2	2.0	2	0.00	0.00	25
	bra4	2	2	2.0	2	0.00	0.00	16
	bra5	2	2	2.0	2	0.00	0.00	25
	cra	2	2	2.0	2	0.00	0.00	25
	par	2	2	2.0	2	0.00	0.00	25
Number of left kineties	bra1	9	5	6.9	7	0.23	5.04	25
	bra2	9	4	6.7	7	1.46	21.79	25
	bra3	8	3	6.2	6	1.36	21.93	25
	bra4	8	6	7.1	7	0.81	11.41	16
	bra5	10	4	6.6	7	1.50	22.73	25
	cra	9	5	6.9	7	0.23	3.28	25
	par	8	4	5.8	6	0.97	16.72	25
Number of basal bodies in a frontoventral kinety	bra1	283	182	225.8	223	5.54	2.45	25
	bra2	293	206	258	253	21.40	8.30	25
	bra3	271	186	217.9	217	26.17	12.01	25
	bra4	353	188	272.2	288.5	50.88	18.69	25
	bra5	261	189	218.3	215	22.79	10.44	25
	cra	131	93	109.0	107	1.92	1.76	25
	par	110	84	95.3	95	7.13	7.48	25
Number of basal bodies in terminal fragment	bra1	17	11	14.0	14	0.29	2.06	25
	bra2	17	9	13.2	13	2.07	15.68	25
	bra3	16	8	12.1	12	2.30	19.01	25
	bra4	17	10	14.1	14.5	2.11	14.96	16
	bra5	15	9	11.6	12	1.66	14.31	25
	cra	7	3	5.5	6	0.17	3.18	25
	par	7	4	5.4	5	0.76	14.07	25
Number of basal bodies in equatorial fragment	bra1	33	8	17.4	14	1.52	8.75	25
	bra2	26	7	15.6	14	5.69	36.47	25
	bra3	22	7	13	13	4.07	31.31	25
	bra4	30	9	20.6	21.5	6.27	30.44	16
	bra5	25	7	13.5	12	4.66	34.57	25
	cra	20	5	11.6	11	1.01	8.72	25
	par	18	4	11.2	12	4.37	39.02	25
Length of macronucleus (μm)	bra1	40	24	33.6	33	0.70	2.10	25
	bra2	59	31	42.96	42	6.60	15.36	25
	bra3	60	23	42.2	43	9.23	21.87	25
	bra4	67	44	53.1	53	6.55	12.34	16
	bra5	75	40	55.6	53	9.64	17.34	25
	cra	27	14	19.9	20	0.54	2.70	25
	par	27	16	23.4	23	2.27	9.70	25
Width of macronucleus (μm)	bra1	19	12	14.8	15	0.32	2.15	25
	bra2	27	17	22.3	22	2.56	11.48	25
	bra3	38	11	23.0	21	7.33	31.87	25
	bra4	32	18	24.4	23	4.32	17.70	16
	bra5	41	19	27.6	27	6.01	21.78	25
	cra	11	6	8.8	9	0.27	3.08	25
	par	19	12	14.9	15	2.27	15.23	25
Diameter of glandule	bra1	10	7	8.4	8	0.13	1.53	25
	bra2	8	4	7.1	7	0.91	12.81	25
	bra3	12	5	7.9	8	1.82	23.04	25
	bra4	11	7	8.6	9	1.15	13.37	16
	bra5	11	7	8.4	8	1.35	16.07	25
	cra	6	3	4.6	5	0.15	3.32	25
	par	6	3	3.6	3	0.81	22.50	25

All data based on protargol-impregnated specimens. Abbreviations: CV, coefficient of variation in %; M, median; Max, maximum; Mean, arithmetic mean; Min, minimum; *n*, number of specimens investigated; SD, standard deviation.

## Data Availability

The data presented in the study are deposited in the NCBI database repository, accession numbers: OL527698, OL527699, OL527700, OL527701, OL527702, OL527703, OL527704.

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
