# Peer review of "A Case Study of the Morphological and Molecular Variation within a Ciliate Genus: Taxonomic Descriptions of Three Dysteria Species (Ciliophora, Cyrtophoria), with the Establishment of a New Species"

_ijms, 2022, doi:10.3390/ijms23031764_

Round 1

Reviewer 1 Report

Presented paper "A case study of the morphological and molecular variation within a ciliate genus: taxonomic descriptions of three Dysteriaspecies (Ciliophora, Cyrtophoria), with the establishment of a new species" is a taxonomical study of an understudied group of ciliates and provide a valuable data concerning their morphological and genetic diversity. However, the description of a new species is not well supported with only a single cell of a single isolate from one geographical location. 

I agree that there is a solid motivation to study the Dysteria genus, with limited knowledge about its genetic diversity and taxonomy. The introduction is insufficient for the broad readership and could be improved. It requires the general overview of ciliates (and their taxonomy) and a more comprehensive introduction to the integrative taxonomy approaches, which are nowadays a gold standard in ciliates and other protists taxonomy. 

The morphological studies of investigated species were performed very well, and in each case, at least 16 cells were analysed per species. On the other hand, molecular analyses were performed on single cells. Single cells might not reflect the SSU rDNA variation within a population. Moreover, a high copy number of SSU rDNA and its variation (https://royalsocietypublishing.org/doi/10.1098/rspb.2017.0425) might be a source of misinterpretations when it comes to the differences in the SSU rDNA sequences, and that was not accounted for in the presented study. Chapter 2.4, referring to the number and positions of differences, is based on a single cell sequence, which might not reflect the diversity of SSU rDNA within a cell. First of all, it has to be made clear within the results section how many cells were analysed per population and species. Second of all cloning procedure should be used to capture the intraspecific variation. I would suggest sequencing more than one cell per analysed species/population if possible.

The phylogenetic analyses are properly performed, but the alignment was manually edited and perhaps trimmed, so it must be submitted to a public repository. 

The new species D. paracrassipes is described, and the morphological and genetic differences to D. crassipes are explained. The sequences of a new species are sister to D. cristata, and one might ask why it is not incorporated into this species. There is one morphological difference listed (number of kineties), but this has to be made clear, why the news species is described. The genetic distance is low, so a very good and stable morphological feature has to be used to describe the new species. Therefore the description of a new species with only a single population analysed and sequence from a single cell are not sufficient.

Is there any knowledge about the geographical distribution of Dysteria species? Currently, vast datasets of environmental data could be analysed to answer this question. Since all new isolates are from a limited geographical region, addressing this question with environmental data would be a valuable addition to our understanding of the diversity of the genus Dysteria

Reviewer 2 Report

In the manuscript entitled “A case study of the morphological and molecular variation within a ciliate genus: taxonomic descriptions of three Dysteria species (Ciliophora, Cyrtophoria), with the establishment of a new species” the Authors improved the knowledge and understanding of the morphological and molecular diversity of the genus Dysteria. To do this, the Authors isolated and characterized three Dysteria species (D. paracrassipes n. sp., D. crassipes, and D. brasiliensis) from subtropical brackish wetlands in China. Moreover, the small subunit ribosomal DNA (SSU rDNA) of each was sequenced and the molecular phylogeny of the genus Dysteria was analyzed. Generally, the MS is well written and structured. Results and discussion are clear and appropriate. However, the M&M section need to be improved. In particular, the section 4.1 related to sample collection. The Authors have to report the methodology used to isolate the species from water. For this reason, I recommended to accept the paper in IJMS after minor revision.
